# Our Experience in Using the Endovascular Therapy in the Management of Hemorrhages in Obstetrics and Gynecology

**DOI:** 10.3390/diagnostics12061436

**Published:** 2022-06-10

**Authors:** Octavian Munteanu, Diana Secara, Maria Narcisa Neamtu, Alexandru Baros, Adela Dimitriade, Bogdan Dorobat, Alexandra Carp, Daniela Elena Gheoca Mutu, Monica Mihaela Cirstoiu

**Affiliations:** 1Department of Anatomy, “Carol Davila” University of Medicine and Pharmacy, 020021 Bucharest, Romania; octav_munteanu@yahoo.com; 2Department of Obstetrics and Gynaecology, University Emergency Hospital Bucharest, 050098 Bucharest, Romania; diachi14@yahoo.com (D.S.); dr_cirstoiumonica@yahoo.com (M.M.C.); 3Department of Obstetrics and Gynaecology, “Carol Davila” University of Medicine and Pharmacy, 020021 Bucharest, Romania; 4Department of Radiology, University Emergency Hospital Bucharest, 050098 Bucharest, Romania; 5Department of Interventional Radiology, University Emergency Hospital Bucharest, 050098 Bucharest, Romania; adela.dimitriade@gmail.com (A.D.); bdorobat@gmail.com (B.D.); alessandra.carp@gmail.com (A.C.); 6Department of Plastic and Reconstructive Surgery, ‘Prof. Dr. Agrippa Ionescu’ Clinical Emergency Hospital, 011356 Bucharest, Romania

**Keywords:** endovascular therapy, vaginal hemorrhage, management

## Abstract

(1) Background: A quarter of maternal deaths are caused by post-partum hemorrhage; hence obstetric bleeding is a significant cause of morbidity and mortality among women. Pelvic artery embolization (PAE) represents a minimally invasive interventional procedure which plays an important role in conservative management of significant bleeding in Obstetrics and Gynecology. The aim of this study was to evaluate the effect and the complications of PAE in patients with significant vaginal bleeding with different obstetrical and gynecological pathologies. (2) Methods: We conducted an observational, retrospective study on 1135 patients who presented to the University Emergency Hospital of Bucharest with vaginal bleeding of various etiology treated with endovascular therapy. All the patients included in the study presented vaginal hemorrhage that was caused by: uterine leiomyomas, genital tract malignancies, ectopic pregnancy, arterio-venous mal-formations and other obstetrical causes. We excluded patients with uncontrolled high blood pressure, severe hepatic impairment, congestive heart failure, renal failure or ventricular arrhythmias. (3) Results: Bleeding was caused in 88.19% of cases by uterine leiomyomas (n = 1001), 7.84% (n = 89) by cervical cancer, 2.29% by ectopic pregnancy (n = 26), 1.23% by arteriovenous malformation (n = 14) and 0.52% by major hemorrhage of obstetrical causes. Endovascular procedures were used in all the cases. In patients with uterine leiomyomas, supra-selective uterine arteries embolization was used. In 97% (n = 1101) of patients, bleeding was stopped after the first attempt of PAE. 3% (n = 34) needed a second embolization. In 12 of 14 cases of AVM, PAE was successful, two other cases needed reintervention; (4) Conclusions: Endovascular procedures represent a major therapy method for both acute and chronic hemorrhage in Obstetrics and Gynecology. It can be used in post-partum or post-traumatic causes of vaginal bleeding, but also in patients with chronic hemorrhage from uterine leiomyomas or inoperable genital malignancies or even as a preoperative adjuvant in cases of voluminous uterine fibroids or invasive malignant tumors, aiming to reduce intraoperative hemorrhage.

## 1. Introduction

A quarter of maternal deaths are caused by post-partum hemorrhage, hence obstetric bleeding being a significant cause of morbidity and mortality among women. Postpartum genital bleeding may appear regardless of whether vaginal delivery or a cesarean section. Bleeding can also be associated with post therapeutic or spontaneous abortion [1]. Other causes of vaginal bleeding are uterine rupture, obstetric trauma, arteriovenous malformation, uterine leiomyomas, and gynecologic malignancies [1,2].

Pelvic artery embolization (PAE) represents a minimally invasive interventional procedure which plays an important role in conservative management of significant bleeding in Obstetrics and Gynecology [2,3]. In literature, pelvic artery embolization (PAE) is shown to have a high rate of success, ranging between 86–90% and 100%, depending on the study [1,2].

Pelvic artery embolization can be used in patients with acute hemorrhage during pregnancy, especially in those who present with placenta accrete and previa [3,4]. Because of vascular collaterals formation during pregnancy, the uterine artery alone might not be the only source of bleeding. Hence, obstruction of the uterine artery alone might not provide hemorrhage control, so visualization of vessels using angiography before embolization should be performed [3].

Ectopic pregnancy is a pathology whose prevalence is increasing, hence the management of patients suffering from it must be improved. Cervical ectopic pregnancy might be a cause of massive vaginal bleeding with life-threatening impact, although it accounts for less than 1% of all ectopic pregnancies. Bilateral uterine arterial embolization in association with curettage decreases the risk of important bleeding and preserves fertility, in comparison with other surgical techniques, such as hysterectomy. PAE can also be used in the management of postpartum hemorrhage, along with symptomatic leiomyomas or cervical cancer [5].

The prevalence of uterine leiomyomas varies between 25% and 80%, being one of the most common benign pelvic tumors in women, which develop between menarche and menopause [6,7]. Although many of the patients are asymptomatic, approximately 25% of them suffer from menorrhagia and secondary iron deficiency anemia, dysmenorrhea, fertility issues, urinary frequency and constipation [6,8]. Submucosal leiomyomas are most associated with vaginal hemorrhage [6]. Besides the fact that PAE is used preoperatively in patients with uterine leiomyomas to reduce hemorrhage during surgery, it can also be used as an individual therapy to cease vaginal bleeding in elected patients with this symptomatology, as an alternative to surgical option [2,6,9].

Retained products of conception (RPOC) represent persistent placental tissue in the uterus, after delivery or abortion. Patients with RPOC may present with symptoms such as genital hemorrhage, abdominal pain and fever. In addition, 18% of RPOC are shown to have marked vascularity on Doppler ultrasound, hence it can be a cause of life-threatening hemorrhage. Kimura et al. claim that uterine artery embolization with gelatin sponge can be safely used in patients with vaginal hemorrhage due to RPOC, considering its clinical success rates and the advantage of preserving fertility. They suggest that microspheres might not be suitable for patients with bleeding caused by RPOC, since spiral arteries in the inter-villous space within the placenta are dilated, having a diameter up to 2 mm. Another reason for which some patients suffer from re-bleeding is ovarian artery supply of RPOC. Thus, they also recommend visualizing ovarian arteries during the initial stages of artery embolization [10].

Arteriovenous malformations (AVM) can be a cause of significant genital bleeding. Acquired AVM develop after gestational trophoblastic disease, curettage for miscarriages and after removal of an intrauterine device [2,11,12]. AVM can also be congenital [2,11]. Bleeding in both forms can be treated with PAE, especially since it has been shown that, after surgical ligation, new collateral vessels form, and hemorrhage may reappear [2].

Premalignant and malignant disorders of trophoblastic tissue are included in the gestational trophoblastic disease (GTD) category. There are cases when high vascular trophoblasts with rapid progression can cause uterine perforation, resulting in a surgical emergency. Hence, PAE should be considered in order to decrease the risk of important hemorrhage. PAE can be used as a preliminary step to surgery but can also be used as an independent therapeutic modality [13].

Vaginal hemorrhage originating from gynecological tumors can be life threatening and can appear during or after treatment [14]. Bleeding occurring from advanced cervical cancer may appear in up to 70% of cases and is the immediate cause of death in 6% of patients with this kind of malignancy [15,16]. In case of emergency intractable bleeding, arterial embolization of branches of the internal iliac arteries represents a promising alternative [14,17,18]. Bilateral embolization should be performed due to collateral pelvic vasculature [14]. Selective artery embolization enables early discharge in these patients [14]. Although more studies are needed to highlight the exact role of PAE in patients with gynecologic malignancies, those already described in the literature demonstrate the promising results of these minimally invasive techniques [2,19].

In addition to its therapeutic role, PAE can be also used as a prophylactic method for preventing bleeding in patients with high hemorrhage risk pathologies, such as ectopic pregnancies [2]. One essential quality is the possibility of uterine preservation and thus of maintaining fertility [1,2,20]. However, the effect of PAE on fertility is debated in the literature, as some studies claim that ovarian dysfunction may appear and miscarriage risk may be slightly increased in patients who underwent this procedure [9,13].

Some of its advantages are the characteristic of being safe, rapid, repeatable, the absence of general anesthesia, avoiding difficulties during surgery, reduced complication rates, short hospital stays, shorter recovery time and earlier return to activities in comparison with other techniques [1,3,6,9]. It also allows the doctor to visualize the origin of the bleeding, thus permitting targeted embolization and preservation of the uterus and its reproductive function [2]. A disadvantage of artery embolization is the occurrence of post-embolization syndrome [1].

Given that this is an interventional technique, this therapy cannot be used in patients with renal insufficiency, coagulopathy or uncorrectable bleeding diathesis, severe hypertension and allergy to contract media [2].

There are some complications of using this kind of procedures, such as angiography related complications (hematoma, pseudoaneurysm or arteriovenous fistula at the site of catheter insertion), infection (such as endometrial or tubo-ovarian abscess, endometritis, urinary tract infection), urinary retention, amenorrhea, uterine infarction and ischemia [2,6,9]. If bilateral internal iliac artery embolization is performed, there may appear complications such as skin necrosis, neurological dysfunction, bladder necrosis, thrombosis and pain [14]. Compared to its relatively low complication rate, PAE represents an effective procedure to cease hemorrhage of obstetrical and gynecological causes [2].

## 2. Materials and Methods

The aim of this study was to evaluate the effect and the complications of PAE in patients with significant vaginal bleeding with different obstetrical and gynecological pathologies.

Between 1 January 2010 and 31 December 2021, we conducted an observational, retrospective, cross-sectional study on 1135 patients who presented to the University Emergency Hospital of Bucharest with vaginal bleeding with various etiology, and who were treated with endovascular therapy. All the patients included in the study presented vaginal hemorrhage that was caused by: uterine leiomyomas, genital tract malignancies, ectopic pregnancy, arterio-venous malformations and other obstetrical causes. We excluded patients with uncontrolled high blood pressure, severe hepatic impairment, congestive heart failure, renal failure or ventricular arrhythmias.

All patients underwent clinical examination along with imaging techniques, in order to achieve a complete overview of the case and to identify the source of the hemorrhage. Vaginal samples and cervical cultures were taken from all the patients included in the study in order to exclude active pelvic infection. Each patient was informed about the procedure, its possible risks and complications and written consent was obtained.

During examination, the most commonly used imaging technique was transvaginal ultrasonography. In limited cases, trans-abdominal ultrasonography and computed tomography were also used. One patient with AVM underwent hysteroscopy, but the examination did not reveal the malformation, which was later found deep in the myometrium. Angiography was used for diagnostic purpose, in order to identify the site of the bleeding, but most importantly it was used as an interventional therapy method. The PAE was carried out through unilateral brachial approach, using a 4F sheath and a cobra diagnostic catheter. During the procedure all patients received heparin sodium (5000 I.U./mL). The materials we used in patients with uterine fibroids, genital tract malignancies or ectopic pregnancy were polyvinyl alcohol particles, microspheres and gelatin sponge. In patients diagnosed with arterio-venous malformations we also used peripherical coils and closure devices. Supra-selective uterine artery embolization was performed in all the patients diagnosed with uterine fibroids. Hence, possible direct anastomoses with the ovarian artery were excluded in these patients.

We were also interested in determining the effect of endovascular therapy on the dysfunctional vaginal bleeding in the group of patients diagnosed with uterine fibroids. During the preoperative period, they were asked to fill a form in order to evaluate their perception of their vaginal bleeding. The patients had to classify dysfunctional metrorrhagia as significant, moderate, minimal or absent. The same questionnaire was filled at follow up visits at 1, 3 and 12 months.

Transvaginal ultrasonography was used in postinterventional control of patients. Using Doppler mode, we were able to evaluate blood flow, and so the disappearance of abnormal vessels which were the origin of the hemorrhage.

## 3. Results

The mean age of the patients included in our study was 41.3 years.

Our study included 1135 patients, 88.19% with uterine leiomyomas (n = 1001), 7.84% (n = 89) with cervical cancer (n = 89), 2.29% with ectopic pregnancy (n = 26), 1.23% with arteriovenous malformation (n = 14), and 0.52% with major hemorrhage of obstetrical causes (n = 5) (see Figure 1).

46.78% (n = 531) of our patients presented with anemia, most of them (n = 521) presenting hypochromic microcytic anemia and only a few (n = 11) presenting chronic anemia associated with malignancy (see Figure 2).

Among 1135 patients, 88.19% presented with uterine leiomyomas (n = 1001), where supra-selective uterine arteries embolization was used. Pre- and postprocedural aspects of uterine artery embolization in a patient with uterine leiomyomas are shown in Figure 1.

7.84% (n = 89) of the patients with cervix cancer needed supraselective embolization of cervicovaginal and long vaginal arteries, as seen in Figure 2 and Figure 3.

Endovascular procedures were used in 2.29% of cases with ectopic pregnancy (n = 26) (see Figure 4 and Figure 5), 1.23% of patients with arteriovenous malformation (n = 14) (see Figure 6), and 0.52% of patients with major hemorrhage of obstetrical causes (n = 6).

In 97% (n = 1101) of patients, bleeding was stopped after the first attempt of PAE. 3% (n = 34) needed a second embolization. In 12 of 14 cases of AVM, PAE was successful, but in the other two cases, reintervention was needed.

Only eight patients out of 1135 stayed in the intensive care unit, with a mean stay of 4.3 days among these cases.

Our complication rate was 7.66% (n = 87). These patients developed a hematoma at the site of catheter insertion. No other complications were recorded.

During preoperative period, the 1001 patients with uterine leiomyomas were asked to fill a form in order to evaluate their perception of their vaginal bleeding. Hence, 80.81% (n = 809) of them perceived dysfunctional metrorrhagia as significant, 11.98% (n = 120) considered it moderate and 7.21% (n = 72) described it as minimal (see Figure 3).

For 1 month follow-up, 904 patients showed up, for 3 months follow-up 512 patients, and for 12 months follow-up only 219. Although some of these did not present to all follow-ups, we were able to observe a decrease in the number of patients complaining of vaginal bleeding and, in patients with persistent hemorrhage, a decrease in the quantity of blood (see Figure 4). Hence, during the 1 month follow-up, which included 904 patients, 2% (n = 18) considered dysfunctional metrorrhagia as significant, 6.52% (n = 59) described it as moderate, 65.15% (n = 589) described it as minimal and 26.33% (n = 238) reported the absence of vaginal bleeding.

During the 3 months follow-up, out of 512 patients, none reported significant metrorrhagia, 2.74% (n = 14) of them described the vaginal bleeding as moderate, 13.47% (n = 69) as minimal and 83.79% (n = 429) as absent. During the 12 months follow-up, when 219 patients showed up, none of them reported significant or moderate metrorrhagia, 20.54% (n = 45) described the bleeding as minimal and 79.46% (n = 174) reported absent vaginal hemorrhage.

## 4. Discussion

To our knowledge, this is the largest study evaluating the success rate and percentage of complications in patients treated with PAE for hemorrhage of obstetric and gynecological etiologies. Our main objective was to correctly assess the incidence of both frequent and uncommon complications that were cited in previous articles, as our study evaluated 1135 patients.

De Brujin et al. claim that postprocedural satisfaction was comparable in patients with symptomatic uterine leiomyomas (symptoms including vaginal bleeding) who underwent hysterectomy versus patients who were treated using PAE. The same study also reports that 69% of women successfully treated by embolization avoided hysterectomy [6,21]. In our study, bleeding originating from uterine leiomyomas stopped immediately after the procedure, but 3% (n = 34) needed reintervention because of hemorrhage recurrence.

Pisco et al. used transcatheter embolization to stop hemorrhage in 108 patients with pelvic malignancies. In 69% of the cases, they obtained complete control of the bleeding, and in 21% of the cases, hemorrhage control was partial [22]. Another study regarding vaginal bleeding originating from gynecologic malignancies is that conducted by Mihmanli et al. [19]. In this study, hemorrhage was stopped in all six patients using polyvinyl alcohol particles [19]. In our study, patients with vaginal bleeding caused by gynecologic malignancies were treated with PAE, also using polyvinyl alcohol particles. Several studies have also shown the effectiveness of internal iliac artery embolization in patients with important vaginal bleeding caused by uterine and cervical cancer [14]. Field et al. claim that selective angiography is mandatory in order to identify the bleeding point and to obtain selective embolization. In cases where the origin of the hemorrhage was not found, they used uterine artery main stem embolization. In all cases, bleeding was stopped immediately [14]. In our study, we used selective artery embolization, occluding both uterine arteries with polyvinyl alcohol particles.

Femoral, axillar, brachial, or radial artery are catheterized during endovascular therapy [9,17]. There are various agents used for embolization, such as metallic coils, glue, spheres, gel-foam particles, gelatin sponge particles, polymer microspheres, polyvinyl alcohol particles or a combination of these [1,9]. Gelatin sponge particle use does not result in permanent embolization, due to degradation [3]. Spheres’ diameters vary in size between 700 and 1200 microns and can be associated with absorbable gel-foam and metallic coils [10,14]. The occluded vessel depends on the size of the used agent. Hence, spheres occlude the capillaries and coils stop the blood flow in more proximal vessels. However, in patients with malignant tumors, occluding larger vessels might promote collaterals development, which will further require reintervention [14]. Zhang et al. claim that all embolic agents are effective, as there is no statistically significant difference between them. However, they do not recommend using coils in patients with arteriovenous malformations and gel-foam in patients with coagulopathy [1]. In our study, the unilateral brachial approach was used in all the patients included. The materials used in patients with uterine fibroids, genital tract malignancies or ectopic pregnancy were polyvinyl alcohol particles, microspheres and gelatin sponge.

Studies conducted by Jacobowitz et al. and by Ghai et al. support the importance of embolization procedures in bleeding caused by AVM, stating that in some of the cases, re-embolization or even hysterectomy might be needed [2,23,24]. In our study, none of the patients needed hysterectomy, but in two cases we had to reintervene and perform another embolization. Barral et al. concluded that PAE in patients with AVM is effective and does not affect fertility. During the study, they used ethylene vinyl alcohol copolymer [25]. In all our 14 patients with AVM, we used gel-foam, peripherical coils and closure devices.

## 5. Conclusions

Endovascular procedures represent a major therapy method for both acute and chronic hemorrhage in Obstetrics and Gynecology. In 97% of patients, bleeding was stopped after the first attempt at PAE. It can be used in post-partum or post-traumatic causes of vaginal bleeding, but also in patients with chronic hemorrhage, for example women with uterine leiomyomas and inoperable genital malignancies. Endovascular procedures can also be used as a preoperative adjuvant in cases of voluminous uterine fibroids or invasive malignant tumors, aiming to reduce intraoperative hemorrhage.

We demonstrated that PAE is very effective in decreasing the abnormal hemorrhage in patients diagnosed with uterine fibroids (12 months after the procedure none of them reported significant or moderate metrorrhagia, 20.54% described the bleeding as minimal and 79.46% reported absent vaginal hemorrhage).

In this study we also observed that this technique has a low incidence of complications in patients that are considered suitable for the procedure given their medical history and the severity of bleeding. The rate of complications is influenced by the pre-interventional evaluation, the technique and the embolization materials used.

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
