# Peer review of "Our Experience in Using the Endovascular Therapy in the Management of Hemorrhages in Obstetrics and Gynecology"

_diagnostics, 2022, doi:10.3390/diagnostics12061436_

Round 1

Reviewer 1 Report

Thank you very much for the invitation to review of the manuscript. It a great pleasure for me.

The purpose of study of Munteanu et al. was to show their experience in using the endovascular therapy in the management of haemorrhages in obstetrics and gynaecology. It is interesting work, however I have a few questions or advice:

1.      These methods have been known for many years, what are the detailed conclusions of this work

2.      I think it is worth precisely defining the goals of this work, both in the text and in the abstract

3.      could the overall health of the patients have had an impact on the success of the procedure? can the authors provide more details on patients?

4.       What were the inclusion and exclusion factors for the study?

5.      Were the patients normal or variant anatomy?

6.      What was the procedural technique?

7.      What were the predictors of failure?

Author Response

Hello and thank you for your time and for your review!

I will answer point by point so everything is organised.

1. Endovascular procedures represent a major therapy method for both acute and chronic hemorrhage in Obstetrics and Gynecology. In 97 % of patients, bleeding was stopped after the first attempt of PAE. It can be used in post-partum or post-traumatic causes of vaginal bleeding, but also in patients with chronic hemorrhage, for example women with uterine leiomyomas and inoperable genital malignancies. Also, endovascular procedures can be used as a preoperative adjuvant in cases of voluminous uterine fibroids or invasive malignant tumors, aiming to reduce intraoperative hemorrhage.

We demonstrated that PAE is very effective in decreasing the abnormal hemorrhage in patients diagnosed with uterine fibroids (12 months after the procedure none of them reported significant or moderate metrorrhagia, 20,54% described the bleeding as minimal and 79,46% reported absent vaginal hemorrhage).

In this study we also observed that this technique has a low incidence of complications in patients that are considered suitable for the procedure given their medical history and the severity of bleeding. The rate of complications is influenced by the pre-interventional evaluation, the technique and the embolization materials that are used.

2. The aim of this study was to evaluate the effect and the complications of PAE in patients with significant vaginal bleeding with different obstetrical and gynecological pathologies.

3+4. All the patients included in the study presented vaginal hemorrhage that was caused by: uterine leiomyomas, genital tract malignancies, ectopic pregnancy, arterio-venous malformations and other obstetrical causes. We excluded patients with: uncontrolled high blood pressure, severe hepatic impairment, congestive heart failure, renal failure or ventricular arrhythmias. Vaginal samples and cervical cultures were taken from all the patients included in the study in order to exclude active pelvic infection.

5+6. The PAE was carried out through unilateral brachial approach, using a 4F sheath and a cobra diagnostic catheter. During the procedure all patients received heparin sodium ( 5000 I.U./ml). The materials we used in patients with uterine fibroids, genital tract malignancies or ectopic pregnancy were polyvinyl alcohol particles, microspheres and gelatin sponge. In patients diagnosed with arterio-venous malformations we also used peripherical coils and closure devices. Supra-selective uterine artery embolization was performed in all the patients diagnosed with uterine fibroids. Hence, possible direct anastomoses with the ovarian artery were excluded in these patients.

7. The rate of complications is influenced by the pre-interventional evaluation, the technique and the embolization materials that are used.

Thank you!

Dr. Daniela-Elena Gheoca Mutu

Reviewer 2 Report

We written and designed study. However it does not anything new . Can the authors suggest any future recommendations ?

Author Response

Hello and thank you for your time and for your review!

In this article, the novelty is the fact that we demonstrated that PAE is very effective in decreasing the abnormal hemorrhage in patients diagnosed with uterine fibroids (12 months after the procedure none of them reported significant or moderate metrorrhagia, 20,54% described the bleeding as minimal and 79,46% reported absent vaginal hemorrhage).

In this study we also observed that this technique has a low incidence of complications in patients that are considered suitable for the procedure given their medical history and the severity of bleeding. The rate of complications is influenced by the pre-interventional evaluation, the technique and the embolization materials that are used. 

Thank you!

Dr. Daniela-Elena Gheoca Mutu